# The structure of the Tad pilus alignment complex reveals a periplasmic conduit for pilus extension

Sasha L. Evans [1,5,9] ✉, Iryna Peretiazhko[1,6,9], Sahil Y. Karnani[2,7], Lindsey S. Marmont[2], James H. R. Wheeler[3,8], Boo Shan Tseng[4], William M. Durham [3], John C. Whitney [2] & Julien R. C. Bergeron [1] ✉

The Tad (Tight adherence) pilus is a bacterial appendage implicated in virulence, cell-cell aggregation, and biofilm formation. Despite its homology to the well-characterised Type IV pilus, the structure and assembly mechanism of the Tad pilus are poorly understood. Here, we investigate the role of the Tad pilus protein RcpC from *Pseudomonas aeruginosa*. Our analyses reveal that RcpC forms a dodecameric periplasmic complex, anchored to the inner membrane by a transmembrane helix, and interacting with the outer membrane secretin RcpA. We use single-particle Cryo-EM to elucidate the structure of the RcpC dodecamer, and cell-based assays to demonstrate that the RcpC-RcpA complex is essential for Tad-mediated cell-cell aggregation. Collectively, these data demonstrate that RcpC forms the Tad pilus alignment complex, which provides a conduit across the periplasm for the Tad pilus filament to access the extracellular milieu. Our experimental data and structure-based model allow us to propose a mechanism for Tad plus assembly.

The multidrug-resistant, Gram-negative bacterium *Pseudomonas aeruginosa* is an opportunistic pathogen implicated in hospital-acquired infections, particularly in immunocompromised patients and those with the genetic condition cystic fibrosis (CF)[1]. *P. aeruginosa* virulence is enhanced through the formation and establishment of biofilms by which the pathogens can colonize and persist on surfaces such as human tissues (e.g., the lungs of CF patients) and indwelling medical devices. Like many other pathogenic bacteria, *P. aeruginosa* encodes multiple secretion systems for the delivery of virulence factors and toxins to host cells, which enhance pathogenesis[2]. Additionally, it possesses several surface appendages such as the flagellum and pili that contribute to increased virulence through their involvement in cell motility, DNA uptake, adherence, and/or biofilm formation[3–6].

The canonical type 4 pilus (T4P), sometimes referred to as the type 4a pilus, is one of the most extensively studied and well-characterized of these appendages. This macromolecular assembly rapidly extends and retracts a long extracellular filament, comprised of thousands of copies of the pilin protein (PilA)[7], through the outer membrane (OM) secretin (PilQ). At the inner membrane (IM), a pilus assembly complex promotes pilin folding and polymerization into a filament, and this process is facilitated by three cytoplasmic ATPases that catalyze extension (PilB) and retraction (PilT and PilU) of the T4P filament[8]. The filament is guided through the periplasm, from the IM assembly complex to the OM secretin, by the alignment complex (PilM, PilN, PilO, and PilP)[9,10].

A related yet distinct type of T4P, the Tad (Tight adherence) pilus, also promotes bacterial virulence through cell-to-cell aggregation,

[1]Randall Centre for Cell and Molecular Biophysics, King's College London, London, UK. [2]Department of Biochemistry and Biomedical Sciences, McMaster University, Hamilton, ON, Canada. [3]School of Mathematical and Physical Sciences, University of Sheffield, Sheffield, UK. [4]School of Life Sciences, University of Nevada Las Vegas, Las Vegas, NV, USA. [5]Present address: Department of Chemistry, King's College London, London, UK. [6]Present address: Francis Crick Institute, London, UK. [7]Present address: Faculty of Pharmacy, University of Toronto, Toronto, ON, Canada. [8]Present address: Department of Clinical Infection, Microbiology and Immunology, University of Liverpool, Liverpool, UK. [9]These authors contributed equally: Sasha L. Evans, Iryna Peretiazhko. ✉e-mail: sasha.1.evans@kcl.ac.uk; julien.bergeron@kcl.ac.uk

colonization of surfaces, and biofilm formation in some bacteria[11,12].The Tad pilus was initially identified in *Aggregatibacter* (*Acinitobacillus*) *acinitomyectemcomitans* clinical isolates that demonstrated a rough colony morphology on solid media, and biofilm formation via auto-aggregation of cells in liquid media[13–15]. Further investigation into the rough colony morphology revealed that these bacteria have long filamentous bundles on their surfaces, comprised of many copies of the Flp (fimbral low-molecular-weight protein) pilin protein[14–17]. Characterization of the locus responsible for this auto-aggregation and biofilm formation led to the identification of a 14-gene locus, subsequently termed the *tad* locus, which has since been identified within a range of both Gram-negative and Gram-positive bacteria, including *P. aeruginosa*[18–21]. Many of the proteins encoded by the *tad* locus have homology to proteins present within canonical T4P systems and the evolutionarily related Type 2 Secretion System (T2SS).

The *tad* locus of *P. aeruginosa* contains 11 of the 14 *tad* genes present within the *A. acinitomyectemcomitans* genome (Supplementary Fig. 1A, B)[21,22]. Like *A. acinitomyectemcomitans*, the filament of the *P. aeruginosa* Tad pilus is comprised of the Flp pilin protein, which is processed into its mature form by a prepilin peptidase, TadV (also denoted FppA) prior to being polymerized into the filament[19,23,24]. In addition to Flp, TadV is also essential for maturation of the TadE and TadF minor pilins, which are also important for Tad pilus functionality[24]. Structure prediction also suggests that TadG functions as a minor pilin; however, this protein has not been characterized to date. TadA, the single cytoplasmic ATPase associated with the Tad pilus, is responsible for driving filament extension via ATP hydrolysis. TadA has been shown to act as a bidirectional extension and retraction ATPase in *Caulobacter crescentus*, but it remains unclear whether this dual function is conserved among all species encoding a Tad pilus[25–27]. TadB and TadC form the assembly complex by coupling ATP hydrolysis to the assembly of processed pilins into the filament. Meanwhile, RcpA is a member of the secretin superfamily of proteins, which are known to form large outer membrane pores through which the pilus can pass to reach the extracellular milieu[28,29]. The lipoprotein pilotin, TadD, is essential for the correct localization and multimerization of the RcpA secretin[28]. TadZ belongs to the ParA/MinD family of ATPases, which are implicated in polar localization of macromolecular assemblies[7,30], although this function has not been reported for the Tad pilus.

It remains unknown if the Tad pilus possesses an alignment complex, because no protein that fulfils this role has been reported to date (Supplementary Fig. 1C), and indeed, none of the Tad pilus proteins have been proposed to be primarily localized to the periplasm. The gene encoding the RcpC protein is conserved among *tad* loci in Gram-negative bacteria, but it exhibits no homology to proteins of known function, and therefore its role has remained poorly understood[21]. Preliminary experiments suggested RcpC localizes to the outer membrane[28], and have implicated it in the glycosylation of Flp pilin proteins[24]. It was also reported that mutational inactivation of *rcpC* does not affect the formation of the Tad pilus in *P. aeruginosa*; however, it compromises the ability for these bacteria to attach to eukaryotic cells[11].

In this study, we present evidence that RcpC is in fact located in the periplasm, anchored to the IM through a single-pass transmembrane (TM) helix. We additionally show that RcpC forms a dodecamer that interacts with the periplasmic region of the outer membrane secretin, RcpA. Functional assays demonstrate that their interaction is necessary for Tad pilus function. Finally, we report the structure of the RcpC dodecamer, which reveals that it forms an aperture with a diameter that is of suitable dimensions to accommodate the Tad pilus filament. Collectively, these data indicate that RcpC forms the alignment complex, joining the IM and OM components of the Tad pilus system and providing a conduit for the Tad pilus filament to extend across the periplasm.

## Results

### RcpC is essential for Tad pilus function

RcpC is conserved across Tad pilus systems, yet its role is poorly understood. To address this, we first sought to test if RcpC is required for overall Tad pilus function. Previous reports indicate that mutational inactivation of the gene encoding the Flp pilin (*Δflp*) does not significantly alter biofilm formation in *P. aeruginosa* PAO1[11,23]. In agreement with this finding, we similarly did not observe a biofilm defect in a *P. aeruginosa* PAO1 strain lacking the *flp* gene (Supplementary Fig. 2A). The authors of these prior studies hypothesized that the plethora of cell surface appendages present on the surface of *P. aeruginosa* may result in apparent functional redundancy in biofilms grown in vitro. In support of this, a role for the Tad pilus in biofilm formation was observed in a *P. aeruginosa* strain lacking both its flagellum and T4P (*ΔfliCΔpilA*)[23]. We attempted to recapitulate these results using the same genetic background but were unable to observe a Tad pilus-dependent decrease in biofilm formation (Supplementary Fig. 2B). We next induced Tad pilus production by overexpressing the transcription activator PprB, which positively regulates Tad pilus gene expression[11], in the *P. aeruginosa* PAO1 *ΔfliCΔpilA* parent strain. Upon inducing *pprB* expression, we observed an increase in Tad pilus protein levels and a corresponding increase in Tad pilus-dependent biofilm formation (Supplementary Fig. 2B, C). Interestingly, this enhanced biofilm formation was accompanied by a striking auto-aggregation phenotype that is characterized by visually apparent floccules in liquid culture and large clusters of bacteria, observable by phase contrast microscopy (Supplementary Fig. 2D, E).

Using these phenotypes as a readout, we next deleted the *rcpC* gene in the PprB over-expressing parent strain (*ΔfliCΔpilAΔrcpC pprB*⁺) to determine if it plays a critical role in Tad pilus function. The resulting *rcpC* deletion mutant exhibited a biofilm and auto-aggregation defect that was comparable to that of the parent strain lacking the *flp* gene (*ΔfliCΔpilAΔflp pprB*⁺; Fig. 1A, B), indicating that it is an essential subunit of the Tad pilus apparatus. Collectively, these results indicate that, although the Tad pilus does not appear to be necessary for wild-type *P. aeruginosa* biofilm formation in these experimental conditions, its overexpression promotes biofilm formation and induces a distinctive cell clumping phenotype. Furthermore, this aggregation phenotype is lost when *rcpC* is mutationally inactivated, demonstrating its necessity for Tad pilus-mediated biofilm formation (Fig. 1B).

### RcpC is an IM protein with a large periplasmic domain

A structural model of RcpC (Supplementary Fig. 3a, b), combined with transmembrane domain analysis of its primary sequence (Supplementary Fig. 3c), predicted that RcpC is comprised of an N-terminal TM helix, followed by two folded domains (D1 and D2) and a C-terminal β-hairpin that localize in the periplasm. Three linker regions, termed L1, L2, and L3, respectively, separate these domains. This domain arrangement suggested that RcpC was likely an IM protein, as OM proteins typically adopt a β-barrel structure[31], contrasting with the previously proposed OM localization of this protein. Within D1, residues present in β-strands β1 and β2, as well as residues 104–112, are conserved across multiple species, whereas the rest of the domain is not well conserved (Supplementary Fig. 3D). By contrast, the majority of D2 is well conserved between species.

To assess RcpC localization, *P. aeruginosa* RcpC was recombinantly expressed in *E. coli*, the cytoplasmic and membrane fractions were isolated, and the presence of RcpC was assessed in each fraction. As shown in Fig. 1C, full-length RcpC (RcpC$_{1-303}$) is predominantly present within the membrane fraction. In contrast, an RcpC construct lacking the predicted N-terminal TM helix (RcpC$_{33-303}$) is predominantly found within the cytoplasmic fraction (Fig. 1C). This confirms that the predicted N-terminal TM helix anchors RcpC to the membrane, whereas the D1-D2 domains are soluble, consistent with RcpC being an IM-anchored single-pass TM protein. Furthermore,

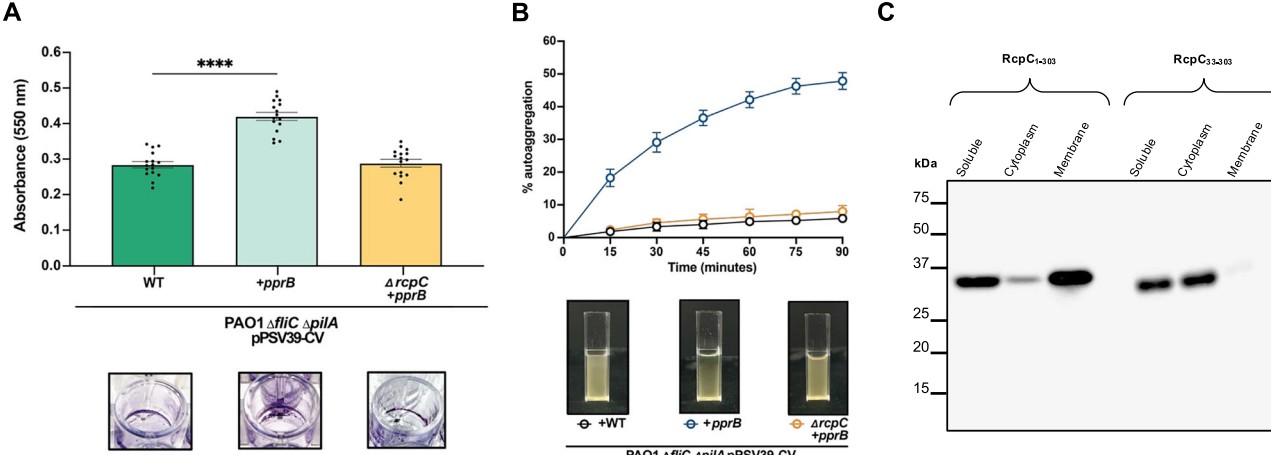

**Fig. 1 | RcpC is an inner-membrane protein essential for Tad pilus formation.**
**A** Crystal violet assay to quantify biofilm formation by *P. aeruginosa* WT strain (left), strain over-expressing the Tad transcription activator PprB (centre), and strain over-expressing the Tad transcription activator PprB but lacking the *rcpC* gene (right). A representative well is shown below. Error bars represent mean values ± SEM, $n = 16$ biological replicates. Asterisks indicate statistically significant differences ($p < 0.05$). **B** Aggregation assay for the *P. aeruginosa* strains as in (**A**). The quantification is shown at the top, and representative images of the corresponding cultures after 90 min are shown below. Error bars represent mean values ± SEM, $n = 5$ biological replicates. **C** Western blot of ultrastructure localization for both full-length RcpC (RcpC$_{1-303}$) (left) and RcpC lacking the predicted N-terminal helix (RcpC$_{33-303}$) (right) recombinantly expressed in *E. coli*. Western blot is representative of three independent repeats.

protein localization prediction for RcpC strongly suggests that the D1-D2 domains are extracellular (Supplementary Fig. 3C). Collectively, these indicate a likely topology for RcpC, with the N-terminal TM helix embedded in the IM, and the D1-D2 domains in the periplasm.

Overall, this finding led us to hypothesize that RcpC may form/be part of the pilus alignment complex, because apart from the periplasmic domains of the RcpA secretin (which are not large enough to span the entire periplasm), no other proteins encoded by the Tad locus are predicted to be localized to the periplasm.

## RcpC forms a trans-periplasmic complex with RcpA

To further characterize RcpC structure and function, we next purified its periplasmic domain (RcpC$_{33-303}$). We noted that during size exclusion chromatography, RcpC$_{33-303}$ eluted earlier than expected based on its predicted monomeric molecular weight, suggesting that it may form a higher-order oligomer. To determine its oligomerisation state, we carried out Size-Exclusion Chromatography coupled with Multi-Angle Light Scattering (SEC-MALS) analysis on RcpC$_{33-303}$ (Fig. 2A). This analysis showed that the predominant RcpC peak had an estimated molecular weight of 365 kDa (±5.61 kDa), which corresponds to 12 copies of RcpC$_{33-303}$. To further verify that RcpC$_{33-303}$ is forming dodecameric complexes, negative stain Electron Microscopy (EM) was carried out on the same sample to directly observe their overall shape (Fig. 2B). The visualization of these particles by TEM revealed that RcpC$_{33-303}$ indeed forms a ring-like oligomeric structure that is ~10 nm in diameter, and contains an apparent pore at its centre.

We next sought to test whether RcpC forms a complex with the OM secretin RcpA to form a trans-envelope conduit[32,33]. To this end, we co-expressed the periplasmic region of RcpC (RcpC$_{33-303}$) with the periplasmic N-domain of RcpA (9.7 kDa)[29], which consists of a small globular Ig domain fold, hereafter referred to as RcpA$_N$. As shown in Fig. 2C, the two proteins co-migrate during gel filtration and elute earlier than RcpC alone, indicating that RcpA$_N$ interacts with the RcpC oligomer. In contrast, RcpA$_N$ in isolation elutes much later (Fig. 2C), demonstrating that in the absence of RcpC, RcpA$_N$ is monomeric.

To further probe the interaction between RcpC$_{33-303}$ and RcpA$_N$, we employed glutaraldehyde crosslinking[34] to identify the formation of a stable complex. As shown in Fig. 2D, bands for the two proteins decrease in intensity upon addition of glutaraldehyde, demonstrating that they form covalently bound higher-order oligomers. Meanwhile, a

clear high molecular weight complex forms at the top of the gel, indicative of the corresponding heterooligomer. This observation confirms that RcpC$_{33-303}$ and RcpA$_N$ form a stable complex.

Finally, we employed SEC-MALS to characterize the RcpC$_{33-303}$-RcpA$_N$ complex. As shown in Fig. 2E, its molecular weight was estimated to be 412 kDa (±2.07 kDa), which is larger than that of RcpC$_{33-303}$ alone (Fig. 2A) and would correspond to a 12:6 RcpC:RcpA stoichiometry. However, we emphasize that this stemmed from proteins expressed on two separate plasmids, and therefore may not reflect the actual stoichiometry of the complex.

Collectively, this data demonstrates that RcpC forms a dodecameric complex that spans the periplasmic space. It also reveals that RcpC interacts with the periplasmic domain of RcpA. This is consistent with the hypothesis that RcpC constitutes the Tad pilus alignment complex, and suggests that it forms a continuous assembly that connects the IM components of the Tad pilus secretion apparatus to the OM-embedded secretin pore.

## RcpC oligomerization and its β-hairpin motif are necessary for its interaction with RcpA

We next sought to gain structural insights into the interaction between RcpC and RcpA. To this end, we employed AlphaFold Multimer[35] to model the RcpA-RcpC complex. As shown in Fig. 3A, this led to a model whereby RcpC interacts with RcpA$_N$ via its C-terminal β-hairpin domain, which completes the RcpA$_N$ β-sheet. This model is of high confidence and contains low positional error for the regions of the model that comprise the RcpA-RcpC interaction (Fig. 3B), indicating high co-evolutionary correlation.

The above experimental and predictive protein-protein interaction analyses for RcpA and RcpC strongly suggest that the RcpC C-terminal β-hairpin motif is required for Tad pilus function. To assess this, we introduced a stop codon into the native *rcpC* gene such that the encoded protein would lack its β-hairpin motif (RcpC$_{1-259}$; see above, Supplementary Fig. 3A). We also introduced a chromosomal VSV-G epitope tag within the linker region connecting the D1 and D2 domains of both full-length RcpC and the strain expressing RcpC$_{1-259}$. Using these strains, we first confirmed by western blot that both forms of RcpC are being expressed to approximately the same levels from their native locus (Supplementary Fig. 4A). We next assessed whether these strains could still form Tad pilus-dependent biofilms and

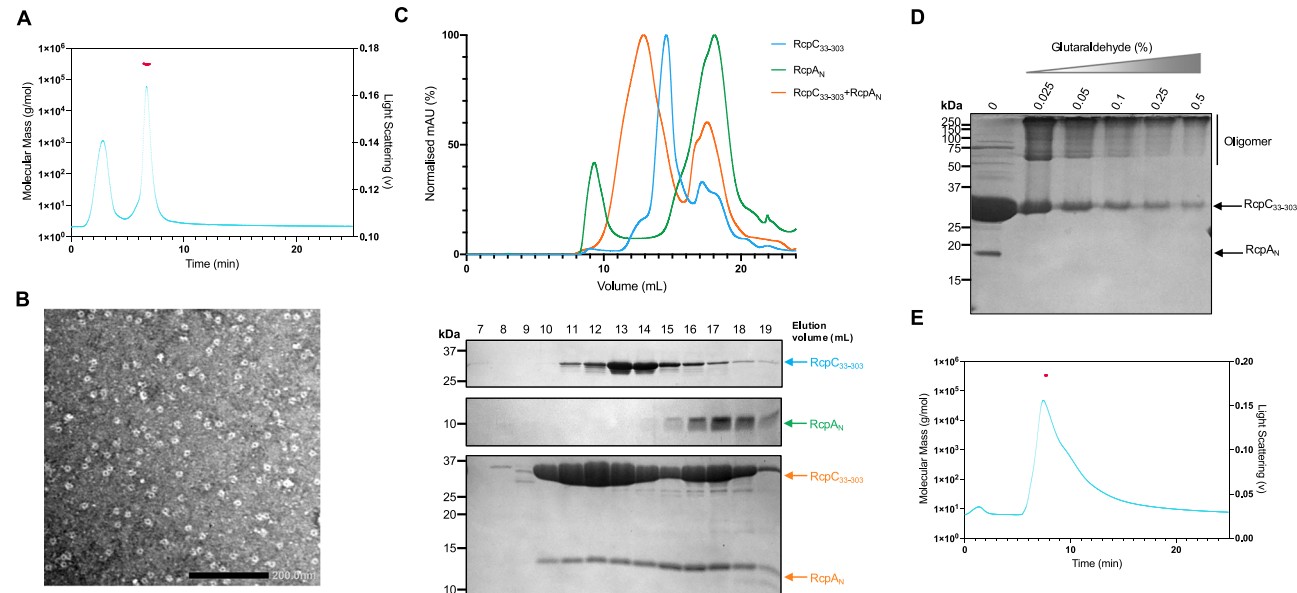

**Fig. 2 | RcpC_{33-303} forms a dodecameric oligomeric assembly that interacts with RcpA. A** SEC-MALS analysis of RcpC_{33-303}. The light scattering trace is in cyan, and the molecular weight of the corresponding peak is in red, revealing a molecular weight of ~ 365 kDa. **B** Negative-stain TEM micrograph of purified RcpC_{33-303}, revealing that it forms ring-like oligomers. Negative stain micrograph is representative of three independent repeats. **C** Gel filtration UV traces for RcpC_{33-303} (Cyan), RcpA_N (Green), or the RcpC_{33-303}-RcpA_N complex (Orange). The corresponding SDS-PAGE gels are shown below. Analytical SEC and subsequent SDS PAGE analysis are representative of three independent repeats. **D** SDS PAGE analysis of glutaraldehyde crosslinking of the RcpC_{33-303}-RcpA_N complex. The band for RcpA_N is absent in the presence of crosslinker, where additional bands of higher molecular weights appearing, corresponding to the crosslinked RcpC_{33-303}-RcpA_N complex. Crosslinking experiment shown here is representative of three independent repeats. **E** SEC-MALS analysis of the RcpC_{33-303}-RcpA_N complex, colored as in (**A**). The molecular weight of the complex is ~412 kDa.

aggregates. As shown in Fig. 3C, only the strain expressing full-length epitope-tagged RcpC forms biofilms in the crystal violet assay, and similarly, the cell aggregation phenotype was not observed in the strain expressing C-terminally truncated RcpC (Fig. 3D). These results demonstrate that the β-hairpin domain of RcpC is necessary for Tad pilus function, likely due to its interaction with RcpA.

We next sought to experimentally show that the β-hairpin domain is necessary for RcpC's interaction with RcpA. To this end, we performed co-purification experiments using a series of RcpC truncations and RcpA_N. As shown in Fig. 3E (top panel), the periplasmic domain of RcpC (RcpC_{33-303}) co-purifies with RcpA_N, which is consistent with our previous co-purification assay (see above, Fig. 2C). Surprisingly, a construct of RcpC lacking the β-hairpin domain (RcpC_{33-249}) still co-purifies with RcpA_N (Fig. 3E, middle panel). This demonstrates that the β-hairpin domain alone is not necessary for the interaction between RcpC and RcpA_N in this assay. In contrast, a construct of RcpC lacking the preceding linker region, L3, and the β-hairpin region (RcpC_{33-233}) did not co-purify with RcpA_N (Fig. 3E, bottom panel), indicating that the L3 linker is necessary for this interaction.

Given that the high-confidence Alphafold model does not predict a direct role for the L3 linker in mediating the RcpC-RcpA interaction, we further characterized the function of the L3 linker using the RcpC_{33-233} construct, which lacks both L3 and the β-hairpin domain. As shown on Supplementary Fig. 4B, size exclusion chromatography indicates that RcpC_{33-233} is monomeric in solution, with SEC-MALS analysis revealing a measured molecular weight of 25.8 kDa (±0.28 kDa) (Supplementary Fig. 4C). These results strongly suggest that L3 is necessary for RcpC oligomerization, and that oligomerization is needed for the interaction between RcpC and RcpA.

### Cryo-EM structural determination of the RcpC periplasmic complex

To provide molecular insight into the oligomeric assembly of RcpC, we employed single-particle cryo-EM to determine its structure

(Supplementary Table 1). Preliminary analysis of purified RcpC_{33-303} revealed large circular particles, consistent with the negative-stain TEM analysis (see above, Fig. 2B), but exhibited preferred orientation, which precluded high-resolution structure determination (Supplementary Fig. 5E). This was mitigated by chemical cross-linking and the addition of N-dodecyl-β-D-maltoside (DDM) to the sample prior to vitrification. A dataset was collected on such sample, where ring-like assemblies exhibiting clear 6-fold symmetry could be readily identified (Supplementary Fig. 5). Additionally, 2D-class averages revealed the presence of additional views of the RcpC complex, indicating that this data is suitable for structure determination. Further processing of the dataset, including imposing 6-fold symmetry, yielded a 2.5 Å resolution Coulomb potential map of the RcpC_{ΔN} dodecamer (Supplementary Figs. 5 and 6A and Supplementary Movie 1). Using this map, we were able to build an atomic model of RcpC_{33-303} containing twelve copies of RcpC (Fig. 4A and Supplementary Table 1 and Supplementary Fig. 6). From this model, we determined that the RcpC dodecamer has a diameter of ~15 nm, and possesses a ~5 nm pore at its centre (Fig. 4A). Remarkably, the Tad pilus filament has previously been reported to have a diameter of ~4.5 nm[36,37], indicating that it is could transit through the RcpC pore.

Interestingly, the D1 domains are arranged into six dimers, whereby each domain only interacts with one other D1 domain (Fig. 4B), whereas the D2 domains are structured as a dodecameric ring with C12 symmetry (Fig. 4C). Accordingly, RcpC adopts two distinct conformations in this structure, which differ in the relative position between D1 and D2 (Fig. 4D), explaining why the overall symmetry of the complex is C6.

Although present within the sample, there is no density for the C-terminal β-hairpin domain, which we could not resolve in spite of extensive 3D classification and masking strategies. However, we observed the presence of diffuse density above the D2 region of the complex in the 2D classes (Supplementary Fig. 5A), which we attribute to this region of RcpC. Additionally, the density for the L3 linker is poorly

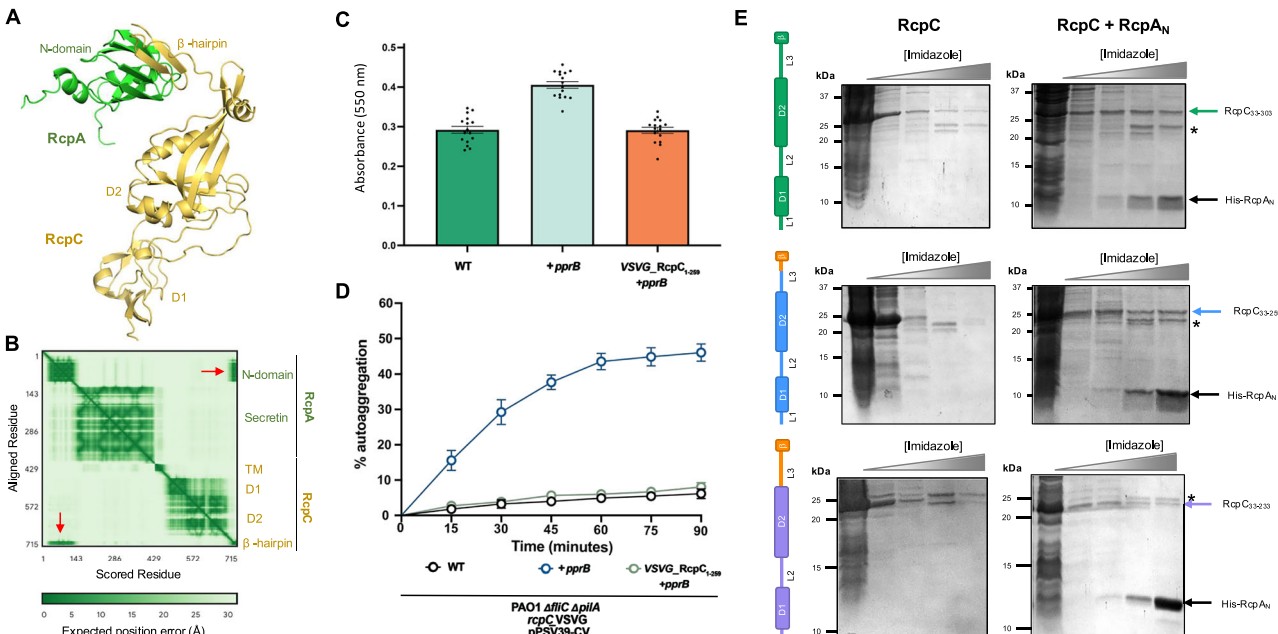

**Fig. 3 | The RcpC β-hairpin, and oligomerization, contribute to the interaction with RcpA. A** AlphaFold Multimer-generated model of the complex formed by RcpA (green) and RcpC (Yellow). **B** Expected position error for the structural model shown in (**A**). Domains of RcpC and RcpA are indicated. The red arrows highlight the interface between RcpCN and the RcpC β-hairpin, which has a distinctive low expected error. **C** Crystal violet assay to quantify biofilm formation by the *P. aeruginosa* ΔfliC ΔpilA strain (left), over-expressing the Tad transcription activator PprB (centre), and strain over-expressing PprB but lacking the C-terminal β-hairpin of RcpC (right). Error bars represent mean values ± SEM, *n* = 16 biological replicates.

**D** Analysis of *P. aeruginosa* auto-aggregation of the strains shown in (**C**), confirming that the RcpC β-hairpin is necessary for Tad-mediated cell-cell interaction. Error bars represent mean values ± SEM, *n* = 5 biological replicates. **E** Co-purification of RcpC$_{33-303}$ (top), RcpC$_{33-259}$ (middle), and RcpC$_{33-233}$ (bottom) with His-RcpA$_N$ (gel on the right), compared to the corresponding purification in the absence of His-RspA$_N$ (left gel). A schematic representation of the corresponding RcpC construct is indicated on the left, with the deleted region in orange. The * symbol indicates an impurity present in the co-purified samples. Images shown are representative of three independent repeats.

resolved, and consequently, we were only able to build residues 233 to 247 and 259 to 268. A 6.5 Å map, obtained from a subset of particles (Supplementary Fig. 7) contained continuous density for the L3 linker, which permitted us to determine the register for residues 259–268. The presence of continuous density for the region between D2 and this region of L3 demonstrates that they form interprotomer interactions with L3 from one protomer interacting with D2 of an adjacent protomer. This observation provides a molecular explanation as to why the deletion of L3 abrogates RcpC oligomerization, as described above.

The overall structure of RcpC demonstrates that individual subunits are twisted around the central pore with the D1 domain of each subunit primarily forming contacts with the D2 domain of the neighboring RcpC subunit (Fig. 5A). Additionally, L3 forms molecular contacts with the D2 domain of the adjacent RcpC molecule (Fig. 5A). These interactions lead to a very extensive interface between RcpC subunits, measuring ~2400 Å² within dimers and ~1930 Å² between dimers, which collectively are comprised of four distinct interfaces (Fig. 5A and Supplementary movie 2). For the D1 dimer interface (interface 1), residues comprising β2 on both subunits form a continuous β-sheet with contacts occurring between Val78, Glu79, and Arg82. Additional contacts stabilizing this interaction are formed between Leu81, Thr83, Pro85, and Ile109 (Fig. 5B). Interface 2 is comprised of contacts between the D1 domain and η4 of the adjacent D2 domain, whereby a hydrophilic loop of D2 is buried within the core of the D1 dimer. This is mediated by residues Ile186, Val188, Asp194, and Arg195 from η4, which form contacts with Ala112, Pro120, and Leu121 from D1 (Fig. 5C). Next, contacts are formed between adjacent D2 domains (interface 3). Most notably, α4 packs against β4 of the neighboring subunit through hydrophobic interactions that involve Val217 and Met221 from α4 and Met155, Phe157 from β4. Additionally, Gln225 from α4 and Arg230 from β7 form a salt bridge that likely stabilizes the complex. Finally, Arg130 from β3 is responsible for

creating a hydrogen bond with Asp153 of β4, which may be important for oligomerisation (Fig. 5D). For interface 4, as indicated above, L3 interacts with D2 from an adjacent subunit, which appears critical for RcpC oligomerisation (See above, and Supplementary Fig. 4B, C). For this interface, the hydrophobic side of the amphipathic helix α7 packs against the hydrophobic regions of β4 and β5. Contacts within this region are primarily mediated by Leu261, Asp264, Leu266, Leu267 and Gln268 residues, which interact with Asp160, Arg162, Leu169, Gln171 and Arg219 from the adjacent D2 domain (Fig. 5E).

## A structural model of the Tad pilus complex

Previous structural studies have revealed the structures of the Flp filament[36,37], the TadA ATPase[26,27], and the RcpA OM secretin[29] in isolation. Combined with the structure of the RcpC alignment complex reported here, these structures provide an almost complete structural model of the Tad pilus export system. We therefore sought to combine these structures to build an atomic model of the full Tad pilus assembly. To this end, we first filled in the missing regions of the available experimental structures, such as the N-domain of RcpA, and the TM helix and β-hairpin of RcpC, using AlphaFold (Supplementary Fig. 8A, C). We also used the reported helical symmetry for the Flp filament to extend its atomic structure beyond the asymmetric unit to generate a model of an extended filament (Supplementary Fig. 8B). We next generated models of the inner membrane components TadB and TadC. Previous modeling attempts of these components using AlphaFold Multimer suggested a stoichiometry of 6:3:3 for the TadABC complex[26]. Using the recently released AlphaFold3 server[38], we obtained an atomic model for this complex (Supplementary Fig. 8D), which agrees with the reported stoichiometry and its associated structure-function analysis. Finally, our functional and molecular data indicate that both the β-hairpin domain and oligomerization are required for RcpC interaction with RcpA$_N$. We therefore attempted to

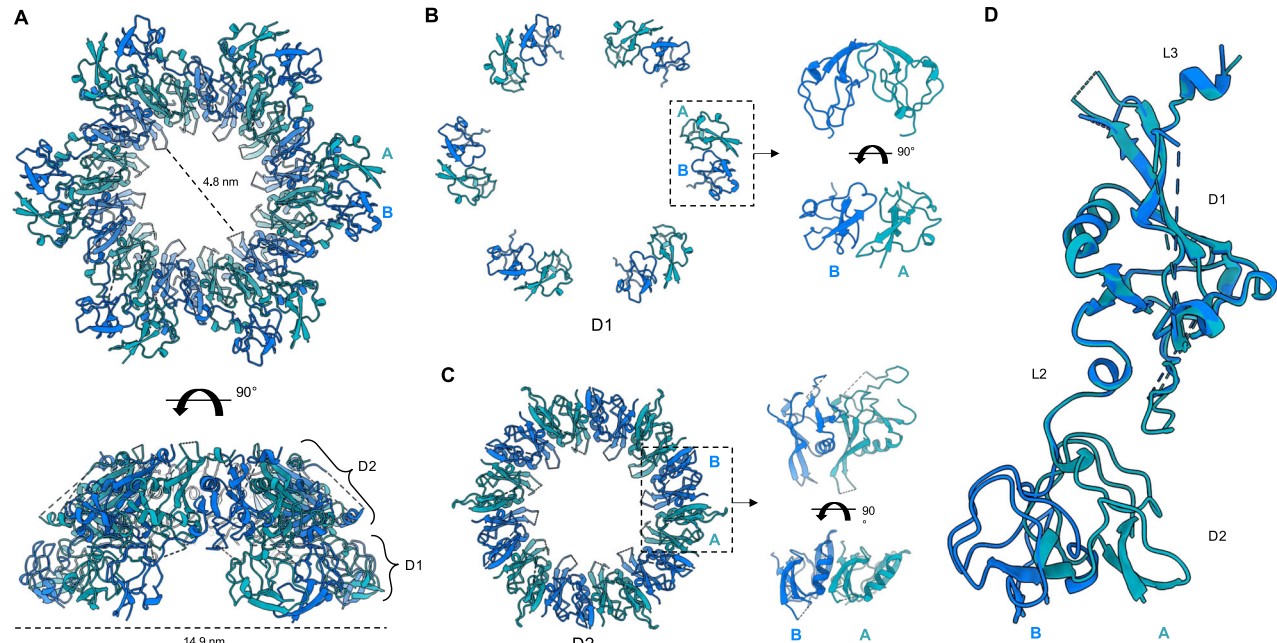

**Fig. 4 | Structure of the RcpC$_{33-303}$ dodecamer. A** Cryo-EM structure of the RcpC$_{33-303}$ dodecamer shown in cartoon representation, with chains alternating between conformation A (Teal) and conformation B (Blue). **B** The structures of the D1 domains **B** and D2 domains **C** are shown in isolation, illustrating their distinct arrangement, consisting of 6 dimers (D1) or a ring with C12 symmetry (D2), respectively. **D** Overlay of the two conformations A, B adopted by RcpC$_{33-303}$ within the RcpC dodecamer, aligned on the D2 domain. The alternate position of D1 relative to D2 is clearly visible.

model the entire RcpA-RcpC oligomer, with the reported 13:12 stoichiometry, using the AlphaFold3 server. As shown on Supplementary Fig. 9A, this led to a model consistent with the aforementioned data whereby RcpA$_N$ is bound to both the β-hairpin and D2 of RcpC at the oligomerization interface. Critically, the positioning error at the RcpA-RcpC interface is low throughout the model (Supplementary Fig. 9B), indicating that this is a high-confidence prediction.

Collectively, this allowed us to propose an atomic model of the intact *P. aeruginosa* Tad pilus complex, spanning both membranes (Supplementary Fig. 8E and Supplementary Data 1). This provides a template for understanding the architecture of this complex and towards establishing the molecular basis for its assembly.

## Discussion

In this study, we show that RcpC is essential for Tad-mediated cell-cell aggregation and that it forms a dodecameric complex in the periplasm that interacts with the outer-membrane secretin RcpA. We also report the structure of the RcpC dodecamer, which adopts a ring-shaped arrangement whose dimensions suggest it guides the Tad pilus filament through its central pore. Finally, we employ a combination of structural modeling, biophysical characterization, and structure-function analyses to show that there are two distinct interaction sites between RcpC and RcpA. Collectively, these results provide compelling evidence that RcpC forms the periplasmic alignment complex for the Tad pilus, providing a continuous channel between the inner membrane assembly complex and the outer-membrane secretin pore.

Recent structural determination of the RcpA-TadD complex demonstrates that RcpA predominantly adopts a 13-mer oligomeric state, although a 14 subunit oligomer was also observed[29]. This, in combination with our RcpC structure, suggests that it may be plausible for this trans-periplasmic complex to adopt a 13:12 stoichiometry. In keeping with this, our modeling data supports a symmetry mismatch between these components. Symmetry mismatch between outer membrane secretins and inner membrane complexes is observed for other secretion systems[39,40], and it is therefore not unexpected for the Tad pilus.

RcpC has no homology to other proteins of known function. However, our data show that it forms the alignment complex in the Tad pilus. Within the T4P and T2SS, PilN/PilO (T4P) and GspL/GspM (T2SS) form heterodimeric alignment complexes that localize to the inner membrane[9,41]. However, our understanding of these alignment complexes is currently limited to low-resolution cryo electron tomography reconstructions[42–46] and crystal structures of the truncated periplasmic domains of PilN, PilO, GspL, and GspM (PDB: 4BHQ, 2RJZ, 5HL8, and 1UV7)[47–49]. Although comparison of RcpC to these isolated alignment complex proteins does not demonstrate any notable similarities (Supplementary Fig. 10), modeling of PilN and PilO into the sub-tomogram averages suggested that 12 of each of these proteins could fit into this density, in line with the RcpC dodecamer[43]. It is also notable that RcpC is the only example to date of an alignment complex protein that spontaneously assembles into its dodecameric state in the absence of other components.

The Tad pilus filament has been reported to facilitate relatively nonspecific adherence to a range of surfaces, such as catheters, epithelial cells, or other bacteria[18,20,23]. The conduit formed by RcpC and RcpA may therefore function, in part, to prevent erroneous filament adherence to the peptidoglycan cell wall within the periplasm. It is noteworthy that, unlike other bacterial appendages[50,51], including the related T4P, no peptidoglycan hydrolases or lytic transglycosylases have been reported in association with the Tad pilus. It is possible that this system recruits PG-cleaving enzymes involved in other cellular processes for this function. If this were the case, we hypothesize that, based on its periplasmic localization, RcpC is the most likely component involved in this recruitment.

The assembly of large appendages spanning both membranes of Gram-negative bacteria involves multiple coordinated interactions of the large number of proteins that assemble each of these systems[52–54]. In the case of the T4P, pilus elongation and retraction are relatively well understood, but the molecular details underlying the formation of the transmembrane complex are not known[8]. Here, we show that RcpC possesses a central pore that closely matches the diameter of the Tad pilus filament, suggesting that it acts as a channel that facilitates the

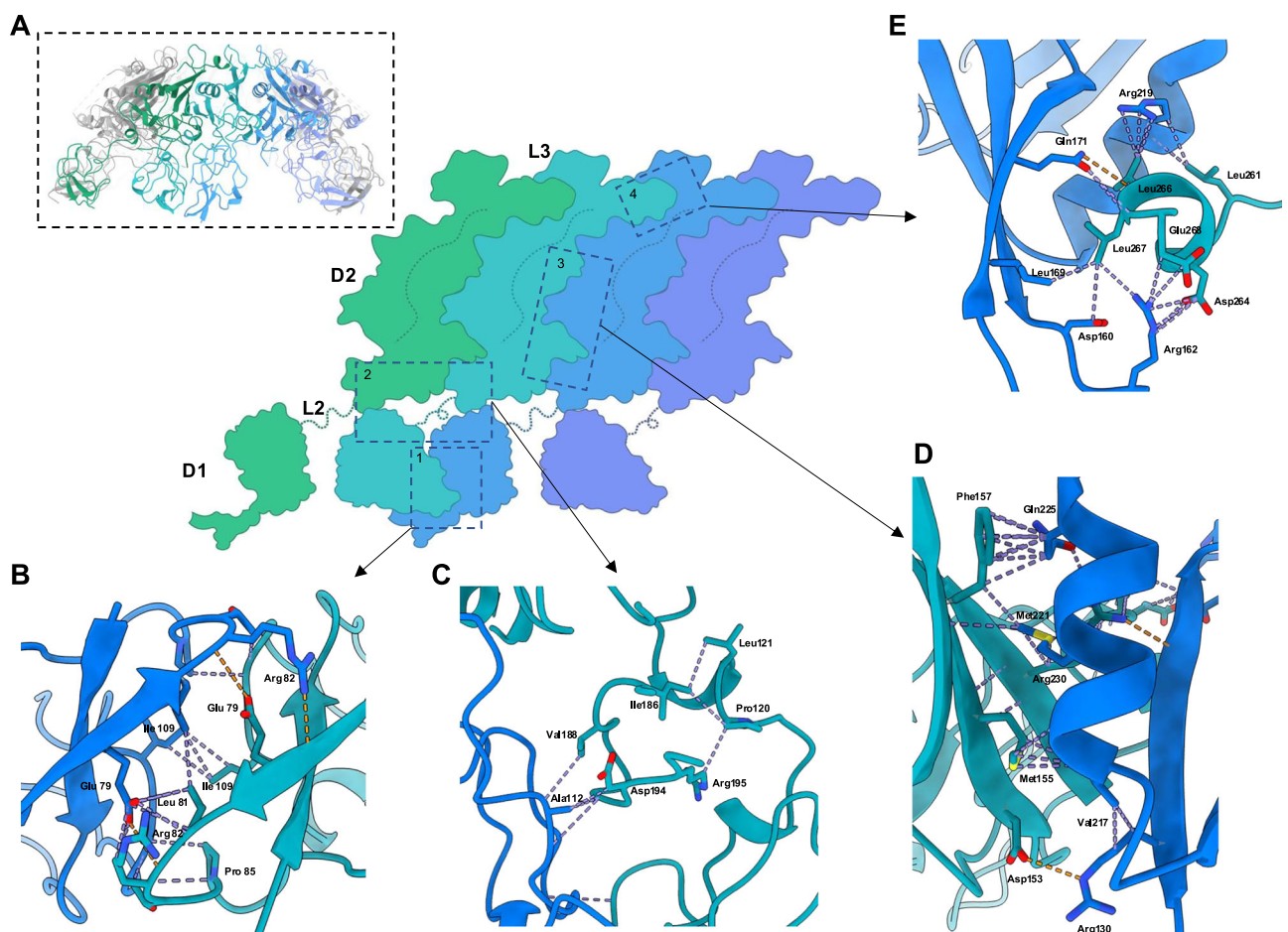

**Fig. 5 | Details of the interactions between Adjacent RcpC molecules in the dodecameric structure. A** Schematic representation of the arrangement of domains for four adjacent molecules within the RcpC$_{33-303}$ dodecamer structure (shown in insert for reference). Detailed views of the four interfaces between adjacent molecules, namely interfaces 1, 2, 3, and 4, are shown in **B**, **C**, **D**, **E**, respectively. VDW contacts are shown in purple, and Hydrogen bonds are in orange.

passage of the filament across the cell envelope. Therefore, filament polymerization at the inner membrane needs to be prevented prior to the formation of the full RcpAC trans-envelope complex, to avoid the filaments from being formed into the periplasmic space instead of outside the cell. We speculate that the assembly of the Tad pilus may be induced by the physical interaction between the RcpA secretin and the RcpC alignment complex. In this model, a sub-complex consisting of RcpC and TadABC would form at the inner membrane, and pilus assembly would either not occur or be arrested within the periplasmic RcpC pore (Fig. 6A), perhaps due to the RcpC β-hairpin region forming a closed lid structure at the top of the RcpC channel. Subsequently, the interaction of the N-domain of RcpA with the β-hairpin region of RcpC (Fig. 6B) would not only result in the recruitment of the secretin complex but also induce a conformational change in the RcpC's β-hairpin region that would enable the passage of the filament through the entire Tad complex (Fig. 6C). Cryo-EM analysis of frozen-hydrated sections demonstrates that the width of the *P. aeruginosa* periplasmic space is ~24 nm[55], whereas our measurement for the periplasmic region of our modeled RcpA:RcpC complex is 21 nm. While there is a small discrepancy between these values, we reason that the linker regions contained within these proteins may be able to account for this. However, further studies will be required to test the validity of this proposed assembly model.

## Methods

### Pseudomonas aeruginosa strain generation
Allelic exchange for genetic knockouts and chromosomal tagging in *Pseudomonas aeruginosa* followed the protocol by Hmelo et al.[56], with

constructs cloned into the pEXG2 plasmid. These plasmids were first transformed into *E. coli* XL-1 Blue cells and then into SM10 cells for conjugation into *P. aeruginosa* PAO1. Genes for overexpression were cloned into the pPSV39-CV plasmid[57] and transformed into *P. aeruginosa* via electroporation[58].

A list of primers and plasmids used in this study, their source, and corresponding antibiotic resistance, is included in the Supplementary Tables 2 and 3.

### Biofilm assay
The crystal violet assay was performed as described previously[23]. Briefly, bacterial cells were cultured overnight in Millipore Sigma Brain Heart Infusion (BHI) broth at 37 °C from frozen bacterial stocks. The overnight culture was diluted to an OD$_{600}$ of 0.07 in fresh BHI medium. We then added 100 µl of bacterial suspension to each well of a polystyrene 96-well plate, which was then sealed and statically incubated at 37 °C for 24 h. Post-incubation, planktonic cells were removed, and the plate was washed with water, then air-dried. The biofilm was stained with 0.1% crystal violet, washed again, and dried overnight. The dye bound to the biofilm was solubilized in 33% acetic acid, and absorbance was measured at 550 nm using a Biotek Epoch microplate spectrophotometer.

### Western blotting
Bacterial cultures were grown overnight in lysogeny broth (LB) at 37 °C. Cultures were adjusted to an optical density (OD) of 0.5, centrifuged at 4000 × g for 10 min at room temperature, and resuspended

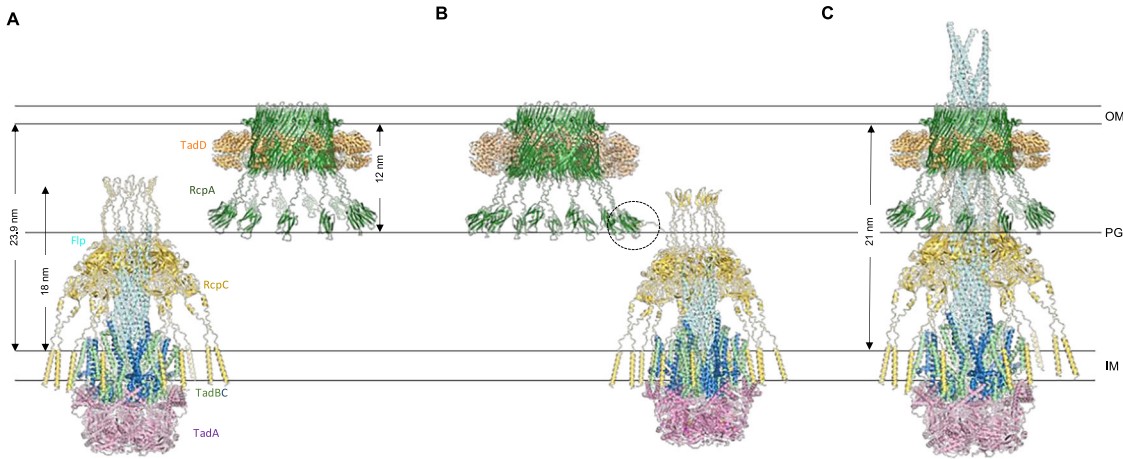

**Fig. 6 | Proposed structural model for the assembly of the Tad pilus. A** Initially, the inner-membrane complex consisting of TadA, TadB, TadC, and RcpA forms. Filament assembly is initiated, but stalls in the periplasm. In parallel, the RcpA secretin complex assembles in the outer membrane. **B** In the periplasm, the RcpA secretin complex is recognized by the RcpC β-hairpin (black dotted circle). **C** Following this interaction, the interaction between the RcpA N-domain and RcpC is completed, forming an intimate complex, allowing the pilus to continue its elongation.

in 4X SDS-loading buffer (3:1 dilution). Samples were boiled at 95 °C for 10 min, then centrifuged at $20,000 \times g$ for 2 min at room temperature. Equal volumes of each sample were run on an SDS-PAGE gel, transferred to a nitrocellulose membrane, which was then blocked with Tris-buffered saline containing 0.05% Tween 20 (TBS-T) and 5% skimmed milk for 30 min at room temperature. It was then incubated with primary antibodies (anti-SV-G antibody, MilliporeSigma, #V4888, 1:5000 dilution; and Anti-RNAP antibody, BioLegend, #663205; 1:5000 dilution) for 1.5 h and secondary antibodies for 1 hour, both at room temperature. After three washes with TBS-T, the membrane was developed using BioRad Clarity Western ECL substrate.

### Auto-aggregation assay

The auto-aggregation assay was performed as previously described[59]. Specifically, PAO1 strains were cultured overnight in Millipore Sigma Brain Heart Infusion (BHI) broth at 37 °C using frozen bacterial stocks. Cells were subsequently diluted to an optical density at 600 nm of 0.07 in fresh BHI medium. One milliliter aliquots of the cultures were incubated in BioRad semi-microvolume cuvettes at room temperature for various timepoints (0 to 90 min, in 15-min increments) and photographed. At each time point, percent auto-aggregation was calculated by subtracting the optical density at the given time point from the initial optical density, dividing this difference by the initial optical density, and multiplying by 100. This value reflects the proportion of cells that aggregated and settled out of suspension over time. The assay was conducted in triplicate, with one representative image shown.

### Phase contrast microscopy

For phase contrast microscopy, frozen bacterial stocks were streaked on 1.5% LB agar plates with Gentamicin at 30 µg/ml. Isolated colonies were cultured in BHI broth with Gentamicin and incubated overnight at 37 °C. Cultures were diluted 1:100, expression of PprB was induced with 1 mM IPTG, and cultures were further grown for 3 h. The cultures were then immobilized on 1.5% LB agarose pads and covered with #1.5 coverslips. Imaging was performed using a 100x oil immersion objective and a Ph3 annulus.

### Protein expression and purification

Genes encoding for protein expression were codon optimized and cloned into pET28a (RcpA) or pET21a (RcpC) (Novagen). Mutants and truncations were engineered by site-directed mutagenesis, using the QuickChange II kit (Agilent). The primers for this are listed in Supplementary Table 2.

For expression of all protein constructs, plasmids were transformed into *E. coli* BL21 (DE3) cells, and transformants were grown in LB medium supplemented with 100 µg mL$^{-1}$ Ampicillin and/or 50 µg mL$^{-1}$ Kanamycin, as required. Protein expression was induced at log-phase by adding 1 mM isopropyl-L-thio-B-D-galactopyranoside (IPTG), for 16 h at 18 °C. Pellets were resuspended into lysis buffer (50 mM HEPES, 150 mL NaCl, 1 mM EDTA, and 1 mM TCEP, pH 8.0, supplemented with 0.1 mg mL$^{-1}$ DNase 1 (Sigma) and an EDTA-free complete protease inhibitor tablet, Roche). Cells were lysed by sonication, and cell debris was removed by centrifugation at $45,000 \times g$ for 30 min at 4 °C. Soluble supernatant fractions were further clarified by filtration using a 0.45 µm PVDF membrane prior to adding 2 mL Ni-NTA resin (Thermo Scientific) pre-equilibrated in wash buffer (50 mM HEPES, 300 mM NaCl, 20 mM Imidazole, and 1 mM TCEP, pH 8.0). Resin was incubated with supernatant for at 4 °C for 1 hour and subsequently transferred to a gravity flow column, and protein was eluted with a gradient of elution buffer (50 mM HEPES, 150 mM NaCl, 500 mM Imidazole, and 1 mM TCEP, pH 8.0). Fractions containing the protein of interest were pooled and concentrated to 500 µL before loading onto a Superose 6 increase 10/300 GL column (Cytiva) in SEC buffer (50 mM HEPES, 150 mM NaCl, 200 mM Arginine, and 1 mM TCEP, pH 8.0). Fractions containing the required protein were combined and concentrated. Ten percent glycerol was added prior to storage at −80 °C until required for further use.

### Protein localization analysis

To determine protein localization for RcpC$_{FL}$ and RcpC$_{33-303}$ samples, soluble supernatant fractions were obtained (as described above) and ultracentrifuged at $210,000 \times g$ (Type 70Ti Rotor, Beckman) to pellet membranes. Membrane pellets were resuspended in buffer to the same volume of the soluble cytoplasmic fraction. Samples for each fraction were analysed by both SDS PAGE and western blot, using Anti-His HRP antibody (Bio-Rad, # MCA5995P, 1:5000 dilution). Identical volumes were loaded to allow comparison of protein concentrations between fractions.

### Negative stain electron microscopy

Sample of RcpC$_{33-303}$ was applied to glow-discharged 200 mesh carbon-coated copper grids (Agar Scientific) and stained with 1% uranyl acetate. Grids were imaged with a Technai T12 Spirit transmission electron microscope equipped with a 2 K Eagle camera.

## SEC-MALS

For SEC MALS analysis, 50 μL samples at a concentration of 2 mg mL$^{-1}$ were injected onto a HPLC (Agilent), equipped with a WTC 015S5 column, coupled with a DAWN Multi-Angle Light Scattering instrument and an Optilab Refractive Index Detector (Wyatt Technology). For analysis, the column, previously equilibrated in SEC buffer (50 mM HEPES, 150 mM NaCl, 200 mM Arginine and 1 mM TCEP, pH 8.0), was run at a flow rate of 0.8 mL min$^{-1}$ for 20 min. All SEC MALS data were analysed using Astra v7 (Wyatt Technology).

## Glutaraldehyde crosslinking

To the co-expressed and co-purified proteins, 0.025% (v/v) glutaraldehyde was added, and the mixture was incubated at room temperature for 6 min. Tris buffer pH 8 was then added to a final concentration of 100 mM. The purity and quality of the sample were checked using SDS-PAGE.

## Cryo-EM data collection, processing and model building

Three microliters aliquots of cross-linked RcpC$_{33-303}$-RcpA$_N$ at a concentration of 1 mg mL$^{-1}$ were supplemented with DDM to a final concentration of 0.01% and applied to 300 mesh, 2/2 μm hole/spacing holey carbon grids (EMR). A Leica EM GP Automatic Plunge Freezer (Leica) was used to plunge freeze the grids, samples were incubated for 10 s prior to blotting for 3 s. Micrographs were collected using EPU software (Thermo Fisher) on a 300 KV Titan Krios microscope (Thermo Fisher) equipped with a K3 camera (Gatan). 14,847 micrograph movies were recorded with a pixel size of 0.85 Å over 40 frames, with a total dose of 45 e$^-$ Å$^{-2}$, at a defocus range of −0.5 to −2.5 μM.

Cryo-EM data processing was carried out using CryoSPARC v 4.5.3[60]. Patch motion correction and patch CTF estimation were used to align frames from micrograph movies and assess CTF values, respectively. Automated particle picking was initially carried out on a subset of 2000 micrographs, resulting in 737,254 particles. 2D classification was carried out, and the best classes were selected for template picking on the entire dataset, resulting in 14,150,712 particles. Four consecutive rounds of 2D classification were conducted on these particles. The classes displaying clear secondary element features, containing 1,607,548 particles, were selected and used for an ab initio reconstruction with 3 classes. The best reconstruction, containing 1,208,354 particles, was subjected to a further round of 2D classification, from which particles from the best classes were selected for Non-Uniform Refinement with C6 symmetry, and the map was sharpened with DeepEMhancer[61]. To build the atomic model for RcpC$_{33-303}$, an initial model was generated using AlphaFold2 (ColabFold)[62,63], and twelve copies were manually placed in their corresponding location in the EM map. The atomic model was subsequently modified and coordinates for water molecules were added using Coot[64] and subjected to real-space refinement in Phenix[65].

## Structural modeling and structure representation

Atomic models of full-length RcpA and RcpC were obtained from the AlphaFold database[66]. Models for the RcpA-RcpC complex and for the TadABC complex were obtained using the AlphaFold3 server[38]. For building composite models (Supplementary Data 1), individual models were localized in suitable relative positions in ChimeraX[67], and chains were copied, combined, and aligned in PyMol[68]. ChimeraX was used for structure representation throughout.

## Reporting summary

Further information on research design is available in the Nature Portfolio Reporting Summary linked to this article.

## Data availability

The cryo-EM map of the RcpC dodecamer has been deposited to the EMDB, with accession number EMD-51732. The corresponding atomic model has been deposited to the PDB, with accession number 9GZR. For the complete TAD pilus atomic model, we have used the following PDB entries: 8ODN, 8U1K. Source data are provided with this paper.

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

## Acknowledgements

This work was supported by a grant from H.F.S.P. to J.R.C.B., J.W., W.D., and B.S.T. (RGY0080/2021), with an S4S supplement to support IP. S.L.E. was supported by a PhD studentship from King's College London. S.Y.K. was supported by an Ontario Graduate Scholarship. J.R.C.B. also acknowledges funding from the BBSRC (project grant BB/R009759/2, and equipment grants BB/V019732/1 and BB/V01966X/1). WMD also acknowledges funding from the BBSRC (BB/R018383/1). Cryo-EM grids were screened at the Imperial College London cryo-EM facility, and data were collected at the LonCEM facility; we acknowledge Paul Simpson and Nora Cronin, respectively, for support. We are grateful to Weilong Zhao and Andrew Beavil for maintenance of the Cryo-EM processing cluster and assistance with data transfer and processing. We thank Google DeepMind (Dhavanthi Hariharan and Richard Green) for providing early access to the AlphaFold3 server.

## Author contributions

S.L.E. performed the characterization of RcpC, purified the RcpC$_{33-303}$ dodecamer, performed its biophysical characterization and initial cryo-EM characterization, built the atomic structure, and generated the structural figures, with support from J.R.C.B. IP performed the RcpC-RcpA co-purification, biophysical characterization of the complex, and determined the RcpC$_{33-303}$ cryo-EM structure, with support from S.L.E. and J.R.C.B. S.Y.K. and L.S.M. engineered the *P. aeruginosa* strains and performed the functional assays, with support from JCW. JRCB performed the molecular modeling experiments. J.H.R.W., B.S.T., and W.M.D. contributed expertise. S.L.E. wrote the abstract and introduction; S.L.E., I.P., and J.R.C.B. wrote the results; S.L.E. and I.P. wrote the discussion; S.L.E., I.P., S.Y.K., and J.R.C.B. wrote the methods; S.L.E., S.Y.K., and J.R.C.B. prepared the figures.

## Competing interests

The authors declare no competing interests.
