## [Transparent Peer Review file · Nature Communications]

The structure of the Tad pilus alignment complex reveals a periplasmic conduit for pilus extension

Corresponding Author: Dr Julien Bergeron

Version 0:

Reviewer comments:

Reviewer #1

(Remarks to the Author)

This manuscript by Evans et al. describes a combination of functional and structural analysis of the *P. aeruginosa* Tad pilus assembly protein RcpC which the authors conclude forms an alignment complex for the Pa flp/tad pilus. I believe this conclusion is well-founded based on the combination of i) the topology of the RcpC oligomeric structure, ii) the localization of FL-RcpC to the membrane through the N-terminus and iii) the requirement of RcpC for tad/flp pilus function. In my opinion the results described here advance the field by presenting a viable model for the assembly of tad/flp pili which has implications for related systems (including T4P and TIIS systems). There are some concerns with how these data are presented, both for functional assays and structural data, detailed below.

- 1) The example images provided of the autoaggregation assay are supportive of the authors' interpretation, but I could not find an explanation of how aggregation was calculated in Figure 1B or Extended Figure 2E or the details of the experiment; (ex. what was the starting O.D. of the cultures?)
- 2) For the localization data (also in Figure 1), the RcpC band does appear to be in the membrane fraction in 1C and the cytoplasmic fraction in 1D, but again there are details missing (what was used for the positive control?) and I think the data would be more convincing if these samples were run on the same gel and blotted together. If differences in expression between the full-length and truncated forms make that impractical, presenting the exactly equivalent gels/blots would also be an improvement.
- 3) The PDB report and extended data give the reader reasonable confidence that the RcpC structure is well resolved overall (and the interface contacts described in the main text appear to be on solid footing). However the C-terminal 10-residue stretch around L3 is somewhat cloudier because there are so many unmodeled residues on each side (in sequence) and it does not appear (from the image in E. Fig 6C) that the density is of sufficient quality to assign these residues based on side-chain density. More detail on how the identity of these residues was determined would help reassure the reader on this point.
- 4) The model of the intracellular components of the Pa Tad/Flp system makes excellent use of the authors novel structure and recent advances from other groups. Because the authors have demonstrated that the N-terminus of RcpA binds directly to RcpC, there appears to be little room for additions that would alter the distance from the IM to the OM and hence, the authors should be able to measure the span of their model, arrive at a figure for the Pa periplasmic intermembrane distance and compare that to experimental measurements. It could be that the authors have already done so and simply didn't feel it important to include in Figure 6, but in this case, I think arriving at a reasonable distance from the model would be a useful finding, just as the authors were reassured by the correspondence between the width of the tad pilus and the diameter of the RcpC pore.

Minor comments:

- Line 2 of page 4: the style guide may require *in vitro* to be italicized
- Line 2 of page 5: based on the methods described, it seems this must be the cytoplasmic fraction rather than periplasmic.
- Line of 11 of page 17; *P. aeruginosa* should be italicized
- Although extended data table 1 generally agrees well with the PDB report, there is some variance (ex. percentage of side-chain rotamer outliers); it would be worthwhile to include what program was used to generate the table.

Reviewer #2

(Remarks to the Author)

In this manuscript, Evans et al. present the possible role and cryoEM structure of RcpC protein from the Tad pilus system of *Pseudomonas aeruginosa* O1 (PAO1). The function of RcpC was previously undetermined despite being conserved among Tad gene loci from multiple bacteria species. The authors have shown that RcpC is involved with cell aggregation promoting biofilm formation. Using combinations of affinity-tag purification and chromatography, authors illustrated that RcpC oligomerizes into a dodecameric form in solution which is further supported by cryoEM structure. The authors proposed that RcpC localized in the periplasm and interacted with secretin RcpA to accommodate Tad pilus. Overall, this manuscript could provide valuable insight into the structural assembly of Tad pilus machinery and its biological role in *P. aeruginosa*. I have some comments and questions for authors before I could support its publication.

Major issues;

Line 1-2, page 5. "...is predominantly found within the periplasmic fraction.

From the results in Figure 1, how do authors know that RcpC localizes in the periplasm? I believe that the results only suggest that without its N-terminal TM-helix, RcpC fails to associate with the membrane and is largely found in the cytoplasm. From these results, there is no clear evidence to conclude that it is localized in the periplasm. Perhaps, more suitable experiments such as fluorescent microscopy should be performed as done with RcpA (PMID 37704603) or utilize fluorophore-conjugated anti-VSV-G.

Minor issue;

Line 36, page 1. "models allow us to propose a mechanism for Tad pilus assembly"

Is this a typo?

Extended Figure 2D.

The numeric value of the scale bar is very small. Perhaps, it should be described in the caption instead.

Line 31, page 2. "The tad locus of *P. aeruginosa* contains 11"

It will be useful for readers to provide the alignment between tad gene clusters among different species, at least those that authors mentioned in Extended Figure 3D.

Line 21, page 4. "Furthermore, this aggregation..."

Is this the authors' data or is this data missing citation ?

Line 26, page 5. "...RcpAN"

It might be useful to show which parts the RcpAN construct corresponds to in Extended Figure 1B. And, what is its size (kDa)?

Figure 2A.

Do authors also have light microscopy images to show as performed in Extended Figure 2D?

Figure 2C and 2D.

The gels should have protein markers included.

Figure 2C.

Size exclusion chromatography of RcpAN shows major peaks eluted at ~8-10 mL of elution volume but I do not see bands in the gel at the bottom. What is this peak?

The co-elution of RcpC/RcpA. The protein concentration from co-expression seems to be overloaded. Have authors also tried to reconstitute the complex and perform gel-filtration from individually purified RcpC and RcpA?

Line 12-15, Page 6. "The stoichiometry 12:6 Which largely consistent with the relative concentration from SDS-PAGE"

From the SDS-PAGE, it is really hard to see the correlation that the authors mentioned. It might be useful to have RcpC alone as a control.

Line 1, Page 7. "Ne next sought..."

Is this a typo?

Line 13-14, Page 8. "...RcpC33-233 is monomeric in solution"

Have authors performed SEC-MALS for RcpC33-259 ? Is it also monomeric? This could provide a clearer support to Figure 2E.

Line 25, Page 8. "Extended Figure S5"

Please omit "S" for consistency.

Line 22-24, Page 8. "But exhibited preferred orientation....."

Do authors also have the representative micrograph to support this?

Can authors also include the angular distribution of particles of the presented map in Extended Figure 5?

Extended Figure 2C.

Should this western blot using anti-VSVG be moved to Extended Figure 4 or perhaps revise the main text? Since the author first mentioned about this VSVG epitope in Extended Fig 4.

Line 3-5, Page 9. "Interestingly, the D1 dominas"

Do you think this is an artifact from the cross-linking?

Line 8-9, Page 9. ".....no density for C-terminal β -hairpin...."

Have authors performed 3D classification and local-refinement with C12 symmetry imposed?

Figure 5A.

It might be useful to provide an inset of overall structure to show where those four subunits in Figure 5A are located.

Line 36, Page 11. ".....13:12 stoichiometry in situ"

How abundant is Tad pilus expressed on the PAO1 cell surface? It could be that the 14-subunit RcpA served as a functional secretin (albeit its low abundance) to accommodate Tad extension/retraction. I do not think it is a good idea to suggest "in situ" since both structures of RcpC and RcpA are purified (in vitro). In the case of type IV pilus secretin PilQ, both 14- and 15-subunit were also solved from cryoEM (PMID 33338410), but 14-subunit seems to be what assembles in situ (PMID 39477930)

Reviewer #3

(Remarks to the Author)

Version 1:

Reviewer comments:

Reviewer #1

(Remarks to the Author)

The authors have addressed all of the concerns I had with this manuscript in revision and I endorse its publication in its current form.

Reviewer #2

(Remarks to the Author)

The authors have addressed all my comments. I have no more concerns.

Reviewer #3

(Remarks to the Author)

Reviewer #1 (Remarks to the Author):

This manuscript by Evans et al. describes a combination of functional and structural analysis of the *P. aeruginosa* Tad pilus assembly protein RcpC which the authors conclude forms an alignment complex for the Pa flp/tad pilus. I believe this conclusion is well-founded based on the combination of i) the topology of the RcpC oligomeric structure, ii) the localization of FL-RcpC to the membrane through the N-terminus and iii) the requirement of RcpC for tad/flp pilus function. In my opinion the results described here advance the field by presenting a viable model for the assembly of tad/flp pili which has implications for related systems (including T4P and TIIS systems). There are some concerns with how these data are presented, both for functional assays and structural data, detailed below.

1) The example images provided of the autoaggregation assay are supportive of the authors' interpretation, but I could not find an explanation of how aggregation was calculated in Figure 1B or Extended Figure 2E or the details of the experiment; (ex. what was the starting O.D. of the cultures?)

We are grateful to this reviewer for their supportive and constructive comments on our manuscript. We agree that the details regarding our auto-aggregation assays require further clarification. We have amended the experimental details in materials and methods to include the starting optical density at 600 nm (Page 15, lines 33-36), as well as how auto-aggregation was calculated (Page 16, lines 2-5)

2) For the localization data (also in Figure 1), the RcpC band does appear to be in the membrane fraction in 1C and the cytoplasmic fraction in 1D, but again there are details missing (what was used for the positive control?) and I think the data would be more convincing if these samples were run on the same gel and blotted together. If differences in expression between the full-length and truncated forms make that impractical, presenting the exactly equivalent gels/blots would also be an improvement.

We agree with this reviewer that our protein localisation analysis would benefit from being ran on the same gel for consistency. To emphasise the shift of RcpC₃₃₋₃₀₃ from the membrane to the cytoplasmic fraction, we have now done this western blot while ensuring equal amounts of these samples were loaded for a clearer comparison of these two proteins. Figure 1C now includes the data for both samples, and the text has been amended accordingly (Page 5 line 3).

3) The PDB report and extended data give the reader reasonable confidence that the RcpC structure is well resolved overall (and the interface contacts described in the main text appear to be on solid footing). However the C-terminal 10-residue stretch around L3 is somewhat cloudier because there are so many unmodeled residues on each side (in sequence) and it does not appear (from the image in E. Fig 6C) that the density is of sufficient quality to assign these residues based on side-chain density. More detail on how the identity of these residues was determined would help reassure the reader on this point.

The quality of the map, combined with the alphafold model, permitted us to unambiguously build the atomic models for residues 233-247, and 259-268. However, there is no density for the region between residues 247 and 259. In particular, this precluded us from identify which chain of the 259-268 region corresponded to which density.

In order to determine the chain register for this region, we used an intermediate map, at lower resolution (~6.5Å), which we obtained from a subset of particles. In this map, the density for

the entire linker (233 to 268) is continuous. This allowed us to assign the correct chain for each of the 259-268 region.

We have included a figure to illustrate this in the supplementary material (Extended figure 7), and have edited the text accordingly (Page 9 lines 12-13).

4) The model of the intracellular components of the Pa Tad/Flp system makes excellent use of the authors novel structure and recent advances from other groups. Because the authors have demonstrated that the N-terminus of RcpA binds directly to RcpC, there appears to be little room for additions that would alter the distance from the IM to the OM and hence, the authors should be able to measure the span of their model, arrive at a figure for the Pa periplasmic intermembrane distance and compare that to experimental measurements. It could be that the authors have already done so and simply didn't feel it important to include in Figure 6, but in this case, I think arriving at a reasonable distance from the model would be a useful finding, just as the authors were reassured by the correspondence between the width of the tad pilus and the diameter of the RcpC pore.

We thank the reviewer for bringing this to our attention. We have modified the Figure 6 by adding values for both RcpA and RcpC and, the periplasmic width of *Pseudomonas aeruginosa*. Although there are discrepancies between the total distance of these proteins and the experimental value for the periplasmic width, we reason that both RcpA and RcpC contain long linker regions which were unable to be resolved prior to or within this study and as such these may allow this complex to span the entire periplasmic space within *P. aeruginosa*. We have added a sentence discussing this in the revised manuscript (page 13 lines 4-8).

Minor comments:

-Line 2 of page 4: the style guide may require *in vitro* to be italicized.

-Line 2 of page 5: based on the methods described, it seems this must be the cytoplasmic fraction rather than periplasmic.

-Line of 11 of page 17; *P. aeruginosa* should be italicized

These have been corrected in the revised manuscript.

-Although extended data table 1 generally agrees well with the PDB report, there is some variance (ex. percentage of side-chain rotamer outliers); it would be worthwhile to include what program was used to generate the table.

Since the submission of this manuscript, we have improved the map by using local B-sharpening and masking using DeepEMhancer (See methods section, page 17 line 4). Accordingly, we have further refined the RcpC atomic model, with revised statistics, included in the revised extended data table 1. These statistics were obtained with Phenix, as stated in the table legend.

Reviewer #2 (Remarks to the Author):

In this manuscript, Evans et al. present the possible role and cryoEM structure of RcpC protein from the Tad pilus system of *Pseudomonas aeruginosa* O1 (PAO1). The function of RcpC was previously undetermined despite being conserved among Tad gene loci from multiple bacteria species. The authors have shown that RcpC is involved with cell aggregation promoting biofilm formation. Using combinations of affinity-tag purification and chromatography, authors illustrated that RcpC oligomerizes into a dodecameric form in solution which is further supported by cryoEM structure. The authors proposed that RcpC localized in the periplasm and interacted with secretin RcpA to accommodate Tad pilus. Overall, this manuscript could provide valuable insight into the structural assembly of Tad pilus machinery and its biological role in *P. aeruginosa*. I have some comments and questions for authors before I could support its publication.

Major issues:

Line 1-2, page 5. "...is predominantly found within the periplasmic fraction. From the results in Figure 1, how do authors know that RcpC localizes in the periplasm?"

We thank this reviewer for their thorough and helpful suggestions on our manuscript. We agree that referring to this fraction as the periplasm instead of the cytoplasm is confusing and we apologise for this oversight, this has been amended in the text (page 5 line 3).

I believe that the results only suggest that without its N-terminal TM-helix, RcpC fails to associate with the membrane and is largely found in the cytoplasm. From these results, there is no clear evidence to conclude that it is localized in the periplasm. Perhaps, more suitable experiments such as fluorescent microscopy should be performed as done with RcpA (PMID 37704603) or utilize fluorophore-conjugated anti-VSV-G.

This reviewer correctly points out that our experimental data demonstrates that the N-terminal helix is necessary and sufficient for membrane localization, but does not provide evidence for the localization of the remaining of the protein.

The evidence for this comes from the computational analysis of RcpC, shown on Extended data figure 3C, which strongly suggests that residues 35-303 are extracellular. Combined with the fact that OM proteins possess (with only a few exceptions) b-barrelled folds, this allowed us to propose a topology where RcpC is anchored in the IM via its TM helix, with the remaining domains localized in the periplasm. We have clarified this in the text, page 5 lines 5-8.

We acknowledge that we have not demonstrated this experimentally here. However, this would require very complex experiments, beyond the scope of this work. Considering the high confidence of the computational modelling discussed above, and the fact that our data demonstrate a direct interaction between RcpA and RcpC (Figure 3E), we do not think that these experiments are necessary.

Minor issue;

Line 36, page 1. "models allow us to propose a mechanism for Tad pilus assembly"
Is this a typo?

The reviewer is correct, this typo has been changed in the text.

Extended Figure 2D.

The numeric value of the scale bar is very small. Perhaps, it should be described in the caption instead.

To make this clearer for the reader we have included this information in the corresponding figure legend.

Line 31, page 2. "The tad locus of *P. aeruginosa* contains 11". It will be useful for readers to provide the alignment between tad gene clusters among different species, at least those that authors mentioned in Extended Figure 3D.

We agree with the reviewer that this would be useful. However, considering the limitation in figure numbers for Nature Communications, for us to include this would require removing another figure panel. Given that multiple excellent reviews exist encompassing the gene organisation of the Tad pilus, we decided against including such figure. We have however, included citations to these reviews within the text for readers with an interest in this area. We hope that this reviewer will find this satisfactory.

Line 21, page 4. "Furthermore, this aggregation...". Is this the authors' data or is this data missing citation ?

We have added a reference to the corresponding figure in the text to clarify that this is our own data.

Line 26, page 5. "...RcpAN". It might be useful to show which parts the RcpAN construct corresponds to in Extended Figure 1B. And, what is its size (kDa)?

There is currently no structure of the N-domain of RcpA, it is not encompassed by the structure shown in Extended Figure 1B due to the flexibility of this domain with respect to the rest of the RcpA. We have added the size of this domain in kDa within the text to make this clearer.

Figure 2A. Do authors also have light microscopy images to show as performed in Extended Figure 2D?

We think this reviewer means Figure 1A here. The light microscopy characterization reported in Extended Figure 2D, was to verify that increased absorbance corresponds to cell-cell aggregation. However this assay is not quantitative, and the Crystal Violet assay used here is therefore a better characterization of this. For this reason, we did not perform the light microscopy analysis for the samples shown in figure 2A.

Figure 2C and 2D. The gels should have protein markers included.

Molecular weights have been added to these figures.

Figure 2C. Size exclusion chromatography of RcpAN shows major peaks eluted at ~8-10 mL of elution volume but I do not see bands in the gel at the bottom. What is this peak?

As we do not observe any protein present when we analyse these fractions by SDS PAGE and RcpAN is poorly stable in the absence of RcpC, we think that it's likely that this peak contains aggregated RcpAN.

The co-elution of RcpC/RcpA. The protein concentration from co-expression seems to be overloaded. Have authors also tried to reconstitute the complex and perform gel-filtration from individually purified RcpC and RcpA?

We agree that for the co-expression RcpC does indeed appear overloaded however due to the difference in RcpC and RcpAN protein concentrations, the smaller size and thus poorer staining for RcpAN, obtaining a gel whereby RcpAN is visible results in RcpC appearing overloaded.

We have previously attempted to reconstitute this complex and assess the binding of this interaction with individually purified RcpC and RcpA using both gel filtration and isothermal calorimetry (ITC), however, as noted above RcpAN is extremely unstable when purified alone, and consequently these experiments proved challenging to complete. We noted that when we attempted this using ITC, visible aggregates were present in the sample after these experiments were conducted. Again, we attributed this to the reduced stability of RcpAN.

Line 12-15, Page 6. "The stoichiometry 12:6 Which largely consistent with the relative concentration from SDS-PAGE". From the SDS-PAGE, it is really hard to see the correlation that the authors mentioned.

We agree with this reviewer that between the different MW between these proteins, and their distinct binding to the Coomassie stain, assessing their relative concentration from the SDS-page is not entirely convincing. Accordingly, we have removed this sentence from the manuscript.

It might be useful to have RcpC alone as a control.

We have a control of RcpC alone which we previously mention in the manuscript but to clarify we have amended the text to make this clearer.

Line 1, Page 7. "Ne next sought...". Is this a typo?

This is indeed a typo, which has been corrected.

Line 13-14, Page 8. "....RcpC33-233 is monomeric in solution". Have authors performed SEC-MALS for RcpC33-259 ? Is it also monomeric? This could provide a clearer support to Figure 2E.

To purify this construct, we would need to re-clone it in the context of the His-tagged RcpC. As the co-purification indicated that RcpC33-259 still binds with RcpA, we did not think that this is a necessary experiment.

Line 25, Page 8. "Extended Figure S5" Please omit "S" for consistency.

This has been corrected in the text.

Line 22-24, Page 8. “But exhibited preferred orientation.....”. Do authors also have the representative micrograph to support this? Can authors also include the angular distribution of particles of the presented map in Extended Figure 5?

We have included the angular distribution of our final 3D reconstruction for RcpC in Extended data figure 5D as requested.

For the preferred orientation, a representative micrograph and selected 2D classes are shown below:

Because of the limitations in figure numbers, we did not include this in the revised manuscript.

Extended Figure 2C. Should this western blot using anti-VSVG be moved to Extended Figure 4 or perhaps revise the main text? Since the author first mentioned about this VSVG epitope in Extended Fig 4.

The anti-VSVG was first used to determine the expression levels of TadD which is under the control of *pprB*. These increased expression levels correspond to the auto-aggregation phenotype that we observe in extended figure 2B, D and E. This is why the corresponding western blot is shown in extended data figure 2.

In the case of extended figure 4A, the anti-VSVG was used to demonstrate that the RcpC constructs complementing our Δ RcpC *Pseudomonas aeruginosa* strain were expressed under the control of *pprB*.

While we agree with this reviewer that having both western blots on the same extended data figure would be logical, to follow the flow of the story we have kept our initial positioning of the two western blots.

Line 3-5, Page 9. “Interestingly, the D1 domains” Do you think this is an artifact from the cross-linking?

We believe that the arrangement of these domains is representative of their structure *in vitro* as we have additional lower resolution maps from previous attempts to study this protein using cryo-EM without crosslinker or detergent which also show this domain arrangement. It is also visible in the 2D classes shown in above. We did not include this data in the revised manuscript because of figure limitations.

Line 8-9, Page 9. “.....no density for C-terminal β -hairpin....” Have authors performed 3D classification and local-refinement with C12 symmetry imposed?

The reviewer makes a very good point here. During data processing we attempted a range of refinements including a masked refinement with C12 symmetry imposed which did not improve our map or provide any additional details. We also attempted masked 3D classifications however, this did not yield any improvement. We attribute this diffuse density to flexibility of the linker between D2 and the C-terminal B-hairpin and reason that it adopts a range of different conformations when not bound to RcpA. We have added a sentence in the manuscript to reflect this (page 9 lines 22-23)

Figure 5A. It might be useful to provide an inset of overall structure to show where those four subunits in Figure 5A are located.

We have added an additional panel to Figure 5A showing the location of these subunits on our RcpC structure.

Line 36, Page 11. “.....13:12 stoichiometry in situ”. How abundant is Tad pilus expressed on the PAO1 cell surface? It could be that the 14-subunit RcpA served as a functional secretin (albeit its low abundance) to accommodate Tad extension/retraction. I do not think it is a good idea to suggest “in situ” since both structures of RcpC and RcpA are purified (in vitro). In the case of type IV pilus secretin PilQ, both 14- and 15-subunit were also solved from cryoEM (PMID 33338410), but 14-subunit seems to be what assembles in situ (PMID 39477930)

We agree with the reviewer that using our *in vitro* data to speculate about the stoichiometry of this complex *in situ* may not be representative due to the dramatically different environment of the cell. We have therefore modified the text to avoid any confusion for the reader (Page 12, lines 19-20).

Reviewer #3 (Remarks to the Author):
